# Exploring Misclassification Information for Fine-Grained Image Classification

**DOI:** 10.3390/s21124176

**Published:** 2021-06-18

**Authors:** Da-Han Wang, Wei Zhou, Jianmin Li, Yun Wu, Shunzhi Zhu

**Affiliations:** 1Fujian Key Laboratory of Pattern Recognition and Image Understanding, Xiamen 361024, China; 2012120701@xmut.edu.cn (W.Z.); lijm@xmut.edu.cn (J.L.); ywu@xmut.edu.cn (Y.W.); szzhu@xmut.edu.cn (S.Z.); 2School of Computer and Information Engineering, Xiamen University of Technology, Xiamen 361024, China

**Keywords:** fine-grained image classification, misclassification information, confusion information, object categorization

## Abstract

Fine-grained image classification is a hot topic that has been widely studied recently. Many fine-grained image classification methods ignore misclassification information, which is important to improve classification accuracy. To make use of misclassification information, in this paper, we propose a novel fine-grained image classification method by exploring the misclassification information (FGMI) of prelearned models. For each class, we harvest the confusion information from several prelearned fine-grained image classification models. For one particular class, we select a number of classes which are likely to be misclassified with this class. The images of selected classes are then used to train classifiers. In this way, we can reduce the influence of irrelevant images to some extent. We use the misclassification information for all the classes by training a number of confusion classifiers. The outputs of these trained classifiers are combined to represent images and produce classifications. To evaluate the effectiveness of the proposed FGMI method, we conduct fine-grained classification experiments on several public image datasets. Experimental results prove the usefulness of the proposed method.

## 1. Introduction

Fine-grained image classification [1,2,3] has drawn much attention in recent years. Fine-grained images are very similar, which make them hard to distinguish. Many efficient fine-grained classification methods have been proposed.

Some researchers tried to improve fine-grained image classification accuracy by designing discriminative classifiers [4,5,6,7,8,9,10,11,12,13,14,15] (e.g., deep convolutional neural networks [9,11,12]); this is achieved by modeling the variations of fine-grained images. Classifiers are often designed on the class level without considering the distinctive characters of a single image. Since fine-grained images often belong to several visually similar subcategories, researchers have also proposed many efficient models by using knowledge from various domains and greatly improved the fine-grained classification performances [16,17,18,19,20,21]. However, domain knowledge is hard to obtain. Besides, domain knowledge is task-dependent and varies from different image datasets.

Other researchers make use of the structure information of images with multiview correlations. Spatial as well as class-level information [22,23,24,25,26,27,28,29,30] is often used with intensive labeling requirements. Automatic detection of objects is also used; although effective, it introduces noisy information, especially when the number of fine-grained classes is large. Since the discriminative power of single view is limited, one natural way is to combine multiview correlations [31,32,33,34,35,36,37,38,39,40,41,42,43,44]. This is achieved by ensuring consistency of different views. Making use of multiview correlations can eventually improve the performance.

Although many models have been proposed with good performance, they do not explore the discriminative information of a single image. In practice, images cannot always be classified correctly. They are often misclassified with several other classes. Besides, misclassification information is eventually not distributed. For example, when we are classifying red flowers of one particular class, the misclassification probability of red flowers of different classes is much larger than white flowers. The classes being misclassified are biased. Misclassification information should also be used to improve the classification performance.

To make use of misclassification information, in this paper, we propose a novel fine-grained image classification method by exploring the misclassification information of images. The proposed method first makes use of prelearned fine-grained image classification models to obtain the misclassification information of images. Instead of using all images for classification, for one particular class, we select a number of classes that are most likely being misclassified with this class. The images of these selected classes and the particular class are then used to train classifiers. The selection and training processes are conducted for each class. As classifiers are trained with different images of varied classes, the outputs of these classifiers cannot be compared for direct classification. We concatenate the outputs of these learned classifiers to form new image representations and use them for classifier training. We evaluate the proposed method on several datasets, and the classification performance proves the usefulness of the proposed method.

The main contributions of the proposed method lie in three aspects:First, we select a subset of images instead of using all the images by exploring the misclassification information for classification. This helps to get rid of noisy information and improve the discriminative power of learned classifiers.Second, we construct new image representations by combining the outputs of classifiers for fine-grained image classification. In this way, we can make use of a number of prelearned models to boost the classification accuracy.Third, the proposed method has good generalization ability by making use of prelearned classification models for misclassification information extraction and classification.

The rest of this paper is organized as follows. We discuss related work in Section 2. Section 3 gives the details of the proposed method. In Section 4, experiments and analyses on several fine-grained image datasets are given. Finally, we conclude in Section 5.

## 2. Related Work

Fine-grained image classification tries to classify a number of subclass images that belong to a particular class (e.g., flower images). The state-of-the-art fine-grained image classification methods could be roughly divided into two schemes. The first scheme tried to design discriminative classifiers on the class level while the second scheme made use of information beyond class-level supervision.

Discriminative classifiers have been widely used to improve classification performance [2,3,4,5,6,7,8,9,10,11,12,13,14,15]. Earlier researchers used local features with support vector machine (SVM). To make use of the spatial layouts of local features, Lazebnik et al. [4] proposed spatial pyramid partitioning of images. Zhang et al. [2] made use of the low-rank constraint to generate general and class-specific codebooks for fine-grained classification. To avoid local feature encoding loss, Yang et al. [5] combined sparse coding and spatial pyramid matching along with linear classifiers while Zhang et al. [6] made use of an exemplar classifier as well as low-rank decomposed features [7,8] for image representation.

With the fast development of deep convolutional neural networks (e.g., AlexNet [9], VGG [11], and ResNet [12]), fine-grained classification performances have been greatly improved. Bilinear convolutional neural networks have also been introduced [3] to model the two-dimension layouts of image pixels. Chai et al. [15] combined segmentation and classifier training for joint classification. Instead of using class-level supervision, Zhang et al. [10] used image-level classifier by hierarchical learning of the structure information. Semantic classifiers [13] were also used for image classification. Although these well-designed classifiers have been proven useful for classification, they were often designed for general images instead of fine-grained images. Since fine-grained images often pose similar appearances, the intrinsic correlations of fine-grained images should be well explored. A number of works have been made. For example, Wah et al. [17] used the correlations of different birds. To determine the location of objects, Zhang et al. [18] combined parts of images with r-cnns; however, automatic detection introduced noisy information. To alleviate this problem, Cui et al. [19] made use of extra human labor to annotate the bounding boxes of objects. To avoid the influences of background areas, Zhang et al. [20] used objectness proposals to both visually and semantically model object, context, and background separately while He et al. [21] also spatially pooled information for classification. Although great improvements have been made, these methods all ignored misclassification information.

Using only class-level supervision is not enough for efficient classification. To alleviate this problem, researchers tried to make use of extra information, e.g., Russakovsky et al. [22] went one step beyond pyramid pooling by using object-centric spatial pooling. Chen et al. [23] contextualized object detection and classification while Angelova and Zhu [26] combined detection, segmentation, and classification into a unified framework. Lin et al. [27] learned important regions automatically from images while Xie et al. [28] made use of the hierarchical information of image parts. Farrell et al. [29] combined volumetric primitives and posed a normalized appearance for classification. Although effective, these methods’ performances decreased when the number of classes increased.

Combination of multiple information could help alleviate the increment of classes to some extent. For example, Torresani et al. [31] used human-labeled information while Yang et al. [32] explored web images to assist with classification. However, this also introduced noisy correlations. Instead of using visual information solely, Farhadi et al. [33] represented images by attributes or semantics. Attributes were manually annotated by experts, which took lots of human labor. To make use of previous learned knowledge for classification, Zhang et al. [36] generated explicitly and implicitly semantic representations [37]. Wei et al. [38] targeted multilabel image classification while Zhang et al. [39] fused semantic information for event recognition. Wu and Ji [40] transferred information from other sources while Zhang et al. [41] shared labels among different views. 3D information was also used for classification in [43]. Ren et al. [44] used region proposal networks for object detection to assist the classification. These methods treated images of the same class as a whole instead of modeling each image separately. Some classes were relatively more similar than other classes. We should treat different classes of images separately.

Many other efficient classification methods [45,46,47,48,49,50,51,52,53,54,55,56,57,58,59,60,61,62,63,64,65,66,67] have also been proposed. Zhang et al. [57] proposed mapping images into subsemantic space instead of only using visual representations. Weak location information was also used [60] to improve classification performance. Part and pose information were used [62,63]. Girshick et al. [66] used the feature hierarchy while Xie et al. [67] leveraged hyperclass correlation. The combination of semantic representation and multiview information were also proven effective for classification [68,69,70,71,72,73,74,75,76,77,78,79,80,81].

## 3. Fine-Grained Image Classification with Misclassification Information

In this section, we give details of the proposed fine-grained image classification method by exploring the misclassification information of images. We first use the misclassification information from prelearned models for misclassified image selection. The selected images are then used to train a number of classifiers. We concatenate the outputs of learned classifiers for new image representations, which are then used for fine-grained image classifications, as shown in Figure 1. Moreover, to further improve the performance, we design several prelearned models with different backbones for image representation and classification, and combine these prelearned models to leverage the advantages of these models.

### 3.1. Exploring the Misclassification Information

We can make use of prelearned models to improve classification performance. Formally, let $x_{n}^{m}$ be the visual features of the *n*-th image used for the *m*-th prelearned model, n=1,…,N,m=1,…,M, *N* is the number of images, *M* is the number of prelearned models, $y_{n}\in R^{C\times 1}$ is the corresponding label, and *C* is the number of classes. Table 1 gives the symbols used in this paper. The prelearned model refers to the classifier, which can be learned using either local features or deep convolutional neural networks. For example, when local feature is used, the prelearned model can be trained using the support vector machine classifier. When using deep convolutional neural networks, the prelearned models refer to various the state-of-the-art networks. For the *m*-th prelearned model $f_{c}^{m}\left(*\right)$ corresponding to the *c*-th class, we use it to predict the classes of images as
(1)$${\hat{y}}_{n,c}^{m}=f_{c}^{m}\left(x_{n}^{m}\right),$$
where ${\hat{y}}_{n,c}^{m}$ is the predicted class for the *n*-th image and the *c*-th dimension using the *m*-th prelearned model, ${\hat{y}}_{n,c}^{m}$ is the *c*-th dimension of y^nm. Ideally, the predicted classes ${\hat{y}}_{n,c}^{m}$ should be the same as their ground truth labels. However, the prelearned models cannot predict all the images correctly in practice. Some images may be confused with different classes. For images of the same class, their predictions scatter over many classes. The misclassification is not evenly distributed. This is because images differ from each other in both semantics and visual appearances. For example, flower images with a similar color and shape are often misclassified with each other. However, the probability of misclassification is low when classifying flowers with different colors and varied shapes. The misclassification information is often discarded by previous models. However, we believe the misclassification information can also be used to improve classification performance.

Suppose for one particular class that images are often misclassified with several classes. We should concentrate on these misclassified classes to mine useful information instead of taking all image classes into consideration. Besides, different prelearned models have varied misclassification information for each class. The classification performance can be improved by jointly modeling this information.

Specially, for each class, we make use of this information by first selecting several classes that are most likely being misclassified with this class. We calculate the class distribution of y^nm for all images of the *c*-th class with *m*-th model and sort it in descending order. Let
(2)$$d_{c}^{m}=\left[d_{c,1}^{m};\ldots ;d_{c,C}^{m}\right]$$
be the sorted class distribution, where
(3)$$d_{c,1}^{m}\geq d_{c,2}^{m}\geq \ldots \geq d_{c,C}^{m}.$$

We select the top K(K<C) classes in which images have mostly been misclassified. We select the classes that correspond to the first *K* dimensions of $d_{c}^{m}$. Images of the *K* classes along with the *c*-th class are then selected to construct a misclassification subset. In this way, we can obtain a subset of images $x_{i,c}^{m},i=1,\ldots ,N_{c}^{m}$ with K+1 classes, where $N_{c}^{m}$ is the number of selected images corresponding to the *c*-th class and *m*-th model. This avoids using too many noisy images. Since images are often misclassified with the *K*-th classes, we can improve the classification performance by separating the K+1 classes of images. Using a subset of easily confused images is more efficient than classifying all the images. It can get rid of some irrelevant images and increase the classification accuracy.

### 3.2. Confusion Information Based Image Representations and Classifications

To make use of the selected K+1 class images, we train K+1 one-vs-all classifiers to separate them. The advantages of using selected images lie in three aspects. First, we can get rid of irrelevant images and concentrate on the classes that are most likely being misclassified. Second, we can use various state-of-the-art image classification methods to improve the classification performance. Third, since the selection and training processes are conducted independently, it can be paralleled to save the computational time and improve the modeling efficiency.

For images corresponding to the *c*-th class with the *m*-th pretrained model, let $g_{c,k}^{m}\left(*\right),k=1,\ldots ,K+1$ be the learned classifiers that separate the K+1 classes of images apart, we can then make use of the predictions for fine-grained image classification. Various efficiently prelearned models can be combined with the proposed method. However, since each $g_{c,k}^{m}\left(*\right)$ is only used to classify the corresponding K+1 classes of images, the predicted values cannot be directly compared.

To predict the class of one testing image, we can use the learned classifiers. The output of one learned classifier indicates the semantic similarity between this testing image and the class with the corresponding classifier. This information can be used for image representations, which has been proven by [6,13,49,68,69,70]. We use the learned classifiers $g_{c,k}^{m}\left(*\right),k=1,\ldots ,K+1,c=1,\ldots ,C$ as new image representations. For each image $x_{n}^{m},n=1,\ldots ,N$, this is achieved by first using each learned classifier $g_{c,k}^{m}\left(*\right)$ to predict its class as
(4)$$h_{c,k,n}^{m}=g_{c,k}^{m}\left(x_{n}^{m}\right).$$

The predicted values are then concatenated as
(5)$$h_{n}^{m}=\left[h_{1,1,n}^{m};..;h_{1,K+1,n}^{m};h_{c,k,n}^{m};..;h_{C,K+1,n}^{m}\right],$$
where $h_{n}^{m}\in R^{(CK+C)\times 1}$. Note that we use the learned classifiers to predict the classes of all the images instead of images that belong to particular classes for two reasons: First, the selected top *K* classes cannot cover all the confused classes, especially when a relatively small *K* is used. Second, an image is predicted *C* times for final classification. This makes the proposed method more robust and effective than a single classifier.

Making use of the new image representations for final classification is quite straightforward. This can be achieved by learning *C* one-vs-all linear classifiers as
(6)$$w_{c}^{m}=argmin_{w_{c}^{m}}\sum_{n=1}^{N}\ell \left(w_{c}^{m}h_{n}^{m},y_{n}^{m}\right)+\alpha {∥w_{c}^{m}∥}_{2}^{2}.$$

∀c=1,…,C. $w_{c}^{m}\in R^{1\times (CK+C)}$ is the classifier parameter to be learned, α is the parameter that controls the influences of the regularization term, $y_{n}^{m}$ is the corresponding binary label of the *n*-th image with the *m*-th prelearned model, ℓ(∗,∗) is the hinge loss function as
(7)$$\ell \left(w_{c}^{m}h_{n}^{m},y_{n}^{m}\right)=max\left(0,1-w_{c}^{m}h_{n}^{m}\times y_{n}^{m}\right).$$

Finally, we predict the classes of images by linearly combining the predicted results using *M* prelearned models as
(8)$${\tilde{y}}_{n,c}=\sum_{m=1}^{M}\lambda_{m,c}\times w_{c}^{m}h_{n}^{m},$$
and assign the testing image with the class that has the largest ${\tilde{y}}_{n,c}$. We set $\lambda_{m,c},m=1,\ldots ,M$ to be the same value that is equal to using the mean of the predicted values for classification.

Algorithm 1 gives the procedures of the proposed fine-grained image classification with the misclassification information method. First, the prelearned classifiers are trained using the training data, then they are used to predict the images of the training data using Equation (Equation 1). Based on the prediction results, the misclassification information and class distribution can be calculated using Equation (Equation 2). For each class, utilizing the selected *K* classes that are most likely to be misclassified or confused with the class, the classifier is trained again. The new image representation is obtained by concatenating the result of the *C* newly trained classifiers, using Equations (Equation 4) and (Equation 5). With the new image representation, the final classifiers are trained by Equations (Equation 6) and (Equation 8). It should be noticed that using Equation (Equation 6), the classification results can be obtained using one type of prelearned model, while using Equation (Equation 8), results of several/all types of prelearned models are combined to obtain the final results, which is expected to improve the performance.
**Algorithm 1** Procedures of the proposed fine-grained image classification with misclassification information method.**Input:**   Training images $x_{n}^{m}$ and labels $y_{n}$, prelearned classifier $f_{c}^{m}$, *K* testing images.**Output:**   The predicted classes of testing images:   **Training phase**1: Predict the classes of images with prelearned classifiers using Equation (Equation 1);2: Calculate the misclassification information using Equation (Equation 2);3: Train misclassification classifiers using Equation (Equation 4);4: Concatenate the results for new image representation using Equations (Equation 4) and (Equation 5);5: Train the final classifiers using Equations (Equation 6)–(Equation 8).   **Testing phase**6: Calculate the misclassification information with prelearned classifiers using Equations (Equation 1) and (Equation 2);7: Concatenate the predicted results of testing images using Equations (Equation 4) and (Equation 5);8: Predict the classes of testing images using Equations (Equation 6) and (Equation 8).9: **return** The predicted classes of testing images.

## 4. Experiments

To evaluate the proposed method (fine-grained classification with misclassification information, FGMI), we conduct fine-grained image classification experiments on the Flower-102 dataset [1], the CUB-200-2011 dataset [17], and the Cars-196 dataset [54]. Figure 2 shows some example images of the three datasets.

### 4.1. Experimental Setup

Both local-feature-based methods and deep convolutional neural network (CNN)-based methods have been widely used for fine-grained image classification. CNN-based methods have greatly improved over local-feature-based methods. The proposed method can be combined with various prelearned models. We first evaluate the proposed FGMI using local features (FGMI-LF) on the Flower-102 dataset. The Flower-102 dataset, the CUB-200-2011 dataset, and the Cars-196 dataset are also used to evaluate the performances of FGMI when combined with various prelearned deep convolutional neural network models (FGMI-CNN).

To extract local features from the Flower-102 dataset, we followed the same procedure as [4] and densely extracted SIFT features, as in [55]. The minimum scale was 16 × 16 pixels with the overlap set to 6 pixels. We used the same local feature encoding strategies as prelearned models. The codebook size was set to 1024. We used the same data splits as provided in [1]. We calculated the classification accuracy for each class. The final performance was evaluated using mean classification accuracy. As per deep convolutional neural network (CNN)-based methods evaluated on the CUB-200-2011 dataset and the Cars-196 dataset, we followed the same experimental setup as the prelearned models to get the trained classifiers [9,11,12]. We used the same type of deep convolutional neural networks for classifier training with the corresponding prelearned model. Mean classification accuracy was used for performance evaluations. We used the reported results of other baseline methods for direct comparison. The baseline models were selected for two reasons: some models are widely used and extended by researchers; other models have achieved state-of-the-art performance on these three datasets.

### 4.2. The Flower-102 Dataset

This dataset has 102 classes of 8189 flower images with predefined train/validate/test split (10/10/rest images). There are different numbers of images per class, ranging from 40 to 258. The scale, pose, and lighting conditions vary between images. Some classes are visually similar and hard to separate.

Table 2 gives the performance comparisons of the proposed method with several baseline models [1,2,14,20,53,54,73]. We give the performances of the proposed method when combined with these baseline models. FGMI-LF-AFC, FGMI-LF-LR-GCC, FGMI-LF-OCB, FGMI-LF-ICAI, FGMI-LF-BR, and FGMI-LF-$S^{3}$R represent the proposed method combined with prelearned AFC, LR-GCC, OCB, ICAC, BR, and $S^{3}$R models, respectively. We also give the performance of FGMI-LF when jointly combined with AFC, LR-GCC, OCB, ICAC, BR, and $S^{3}$R for classification (FGMI-LF-Combined). We have three conclusions from Table 2 when local-feature-based methods are combined. First, the proposed method is able to improve over these baseline models. This is because we concentrate on the easily confused classes of these prelearned models. Second, the performances vary for FGMI-LF-AFC, FGMI-LF-LR-GCC, FGMI-LF-OCB, FGMI-LF-ICAI, FGMI-LF-BR, and FGMI-LF-$S^{3}$R because the discriminative power of these prelearned models are different. The performances can be improved by making use of other information (OCB) or measuring the similarities of images finely (BR and $S^{3}$R), rather than simply using the training images with histogram similarities. The performances of these baseline models can be further boosted using the proposed method. Third, the performances can be improved by combining these models (FGMI-LF-Combined). The experimental results on the Flower-102 dataset show the effectiveness of the proposed method when combined with local-feature-based models.

We also evaluate the proposed method when combined with two deep convolutional neural networks: ResNet-50 and ResNet-101, abbreviated as FGMI-CNN-ResNet-50 and FGMI-CNN-ResNet-101, respectively. We can see that the proposed FGMI method is able to improve classification performances. Finally, we combine both local feature based methods and deep-convolutional-neural-network-based methods (FGMI-LF-CNN-Combined) for classification. This eventually improves the classification accuracy.

### 4.3. The CUB-200-2011 Dataset

The CUB-200-2011 dataset has 200 different birds of 11,788 images. The images are divided into 5994 training images and 5794 testing images. The images are also labeled with bird locations along with class information. We only use the class information of the images.

We give the performances of the proposed method and other models [3,11,18,19,60,61,62,63,64,65,71,72,74,75,76,77,78] on the CUB-200-2011 dataset in Table 3. One thing needs to mention is that some baseline models use both bounding box information and image labels. The proposed method only uses image labels. Most baseline models use AlexNet [9], VGG [11], and GoogleNet [58] with variations. Hence, we combine the proposed method with AlexNet, VGG, GoogleNet, and Bilinear CNN (FGMI-CNN-AlexNet, FGMI-CNN-VGG, FGMI-CNN-GoogleNet, and FGMI-CNN-BCNN). We also give the combined performances (FGMI-CNN-Combined) by jointly using the prelearned AlexNet, VGG, GoogleNet, and Bilinear CNN models. We can have similar conclusions as above. The consideration of misclassification information is very effective to boost the classification performances. Take FGMI-CNN-AlexNet, FGMI-CNN-VGG, FGMI-CNN-GoogleNet, and FGMI-CNN-BCNN for example, they are able to greatly improve over AlexNet, VGG [11], GoogleNet, and Bilinear CNN. Besides, performances can be further improved by using more discriminative models. The relative improvements decrease with more discriminative prelearned models. Moreover, we are able to outperform other baseline models when combining the four models in a unified way. Especially, we are able to improve over [75,77,78], which make use of ResNet or its variants for fine-grained classification. Once again, this shows that misclassification information is very useful for accurate fine-grained image classification.

### 4.4. The Cars-196 Dataset

There are 196 classes of 16,185 images in the Cars-196 dataset. Images are divided into 8144 training images and 8041 testing images, respectively. The image labels and bounding box annotations are also provided. We only use the class information of images, as on the CUB-200-2011 dataset.

Performances of the proposed method and comparison with other baseline models are given in Table 4. To be consistent with the experimental setup as on the CUB-200-2011 dataset, we also give the performances of FGMI-CNN-AlexNet, FGMI-CNN-VGG, FGMI-CNN-GoogleNet, and FGMI-CNN-BCNN along with FGMI-CNN-Combined. We can see from Table 4 that the proposed method is able to improve over many baseline models [3,11,60,65,66,67,75,76,77]. Particularly, by using misclassification information, we can improve performance over AlexNet, VGG, GoogleNet, and BCNN, respectively. Besides, we are able to improve over [77], which makes use of the structural information of image regions to assist network construction. When analyzing the proposed method’s performances on different datasets, we find that the Cars-196 dataset is relatively easier to classify than the CUB-200-2011 dataset. This is because cars are rigid objects while birds are nonrigid objects. Rigid objects are relatively easier to classify than nonrigid objects. However, by taking the misclassification information into consideration, we can consistently improve classification performance.

### 4.5. Influences of Parameters

The selected number of classes *K* influences the discriminative power of the new image representations. If we set *K* to 1, the proposed method will equal to only using the most easily confused class. All the classes will be used if we set *K* to *C*. To show the influence of *K*, we plot the performance changes with *K* on the Flower-102 dataset, the CUB-200-2011 dataset, and the Cars-196 dataset in Figure 3. We can see from Figure 3 that setting K/C to 0.1∼0.2 is able to obtain satisfactory performances.

α controls the influences of the regularization term in Equation (Equation 6). We plot its influences on the three datasets in Figure 4. We can see from Figure 4 that α should not be too big or too small. If α is too small, it will have very little influence. However, if we set α to a very large value, the optimization of Equation (Equation 6) will be degenerated. Setting α to 0.1∼10 seems to be a better choice, as shown in Figure 4.

The misclassification information also plays an important role for efficient classification. If we do not use the misclassification information, the proposed method will simply equal the combination of prelearned models with averaged predictions. We give the influences of the misclassification information in Table 5 (no CI). We can see that the misclassification information is very useful for classification.

The new image representation scheme is also necessary for accurate classification. This is because the classifier outputs of different subsets cannot be compared directly. One alternative way is to predict the image’s class by voting. This can be achieved by using the predicted classes (instead of the values) over all the selected subsets corresponding to the pretrained models. We give the performances without using the new image representation scheme on the three datasets in Table 5 (no NIR). Since different subsets contain images of varied classes, the performances of this strategy are not as good as the proposed method.

## 5. Conclusions

In this paper, we proposed an efficient fine-grained image classification method by making use of the misclassification information of prelearned models, which has been generally ignored by previous methods. We used the learned classifiers to select misclassified images for each class. The selected images were then used to train misclassification classifiers. The selection and training process were conducted for each class. We combined the outputs of these learned classifiers for new image representations and trained classifiers for final predictions. The misclassification information contains discriminative features that are important for classification of similar classes with similar semantic and visual appearances. Specifically, for the fine-grained classification task, training the classifiers with misclassification information can better extract confused features, which is useful for discriminating similar classes. To evaluate the proposed method’s effectiveness, we conducted fine-grained classification experiments on three fine-grained image datasets. Experiential results and analysis proved the effectiveness and usefulness of the proposed method.

## Figures and Tables

**Figure 1 sensors-21-04176-f001:**
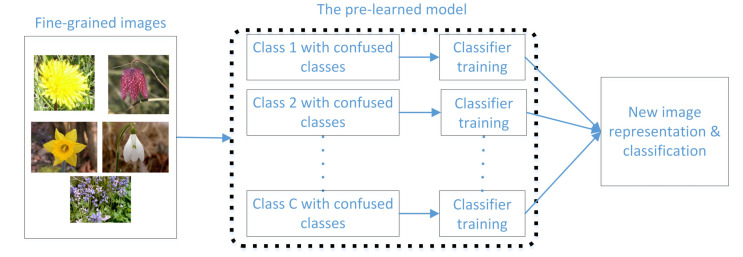
Flowchart of the proposed fine-grained image classification method by exploring the misclassification information.

**Figure 2 sensors-21-04176-f002:**
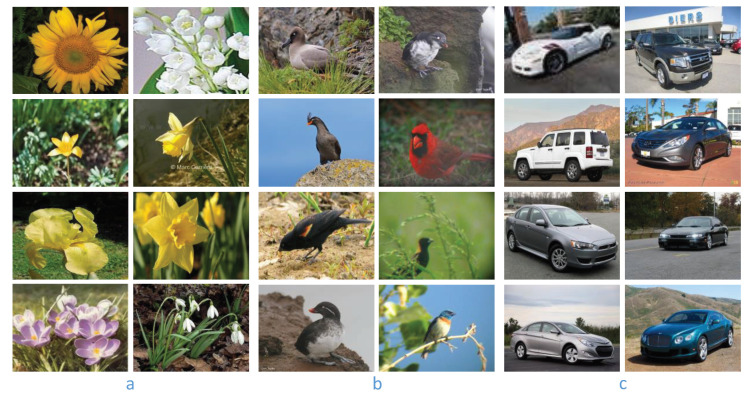
Example images of (**a**) the Flower-102 dataset, (**b**) the CUB-200-2011 dataset, and (**c**) the Cars-196 dataset.

**Figure 3 sensors-21-04176-f003:**
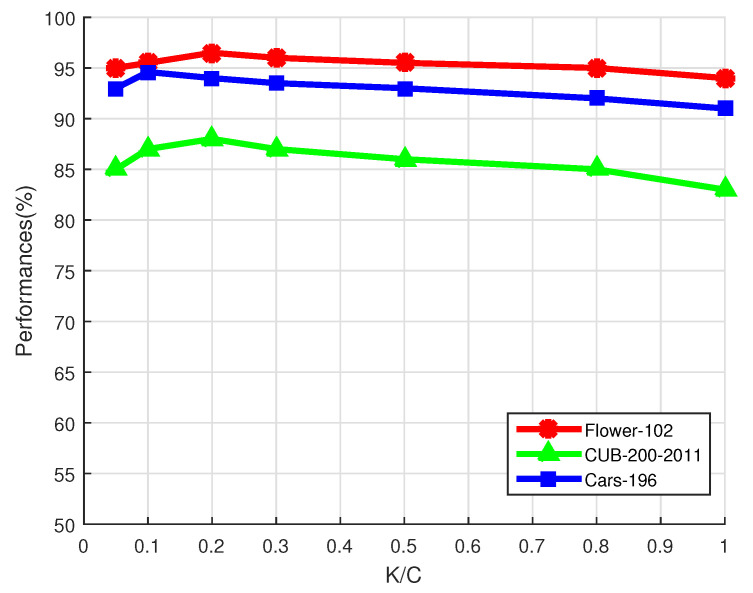
Influence of K on the Flower-102 dataset, the CUB-200-2011 dataset, and the Cars-196 dataset.

**Figure 4 sensors-21-04176-f004:**
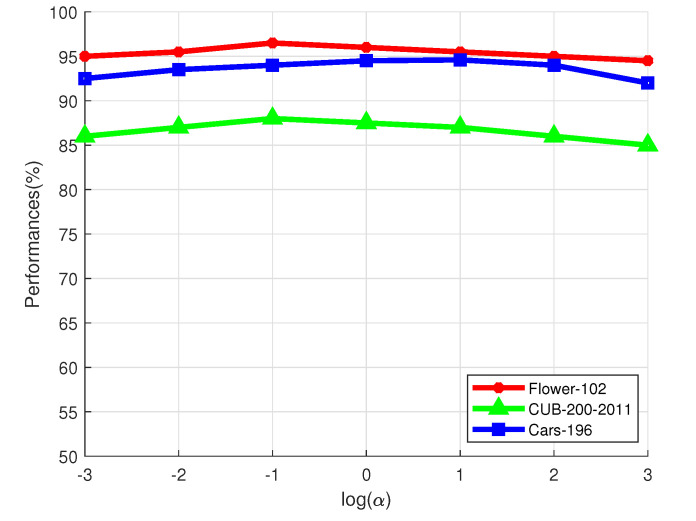
Influence of α on the Flower-102 dataset, CUB-200-2011 dataset, and Cars-196 dataset.

**Table 1 sensors-21-04176-t001:** The symbols used in this paper and their meanings.

Symbol	Description
$x_{n}^{m}$	visual features of the *n*-th image used for the *m*-th prelearned model
*N*	number of images
*M*	number of prelearned models
$y_{n}\in R^{C\times 1}$	label of $x_{n}^{m}$
*C*	number of classes
$f_{c}^{m}\left(*\right)$	*m*-th prelearned model with the *c*-th class
${\hat{y}}_{n}^{m}$	predicted class for the *n*-th image using the *m*-th model
${\hat{y}}_{n,c}^{m}$	*c*-th dimension of ${\hat{y}}_{n}^{m}$
$y_{n,c}$	*c*-th dimension of $y_{n}$
$d_{c}^{m}=\left[d_{c,1};\ldots ;d_{c,C}\right]$	sorted class distribution of ${\hat{y}}_{n}^{m}$ for all the images of the *c*-th class with *m*-th model
*K*	number of selected classes
$x_{i,c}^{m},i=1,\ldots ,N_{c}^{m}$	selected subset of images corresponding to the *c*-th class and *m*-th model
$N_{c}^{m}$	number of selected images corresponding to the *c*-th class and *m*-th model
$g_{c,k}^{m}\left(*\right)$	learned classifier corresponding to the *c*-th class with the *m*-th pretrained model
$h_{c,k,n}^{m}$	predicted value of $x_{n}^{m}$ using $g_{c,k}^{m}\left(*\right)$
$h_{n}^{m}$	concatenated new representation of $x_{n}^{m}$ with the *m*-th model
$w_{c}^{m}$	linear classifier parameter
α	parameter for controlling the influence of the regularization term
$y_{n}^{m}$	binary label of the *n*-th image with the *m*-th prelearned model
ℓ(∗,∗)	hinge loss function
$\lambda_{m,c}$	linear combination parameter

**Table 2 sensors-21-04176-t002:** Fine-grained classification performances of FGMI-LF and other baseline methods on the Flower-102 dataset. FGMI-LF-AFC, FGMI-LF-LR-GCC, FGMI-LF-OCB, FGMI-LF-ICAC, FGMI-LF-BR, and FGMI-LF-$S^{3}$R represent the proposed method combined with prelearned AFC, LR-GCC, OCB, ICAC, BR, and $S^{3}$R models, respectively. FGMI-CNN-ResNet-50 and FGMI-CNN-ResNet-101 represent the proposed method combined with prelearned ResNet-50 and ResNet-101, respectively. FGMI-LF-Combined represents the proposed method combined with AFC, LR-GCC, OCB, ICAC, BR, and $S^{3}$R jointly. FGMI-CNN-Combined represents the proposed method combined with ResNet-50 and ResNet-101 jointly. FGMI-LF-CNN-Combined represents the proposed method combined with FGMI-LF-Combined and FGMI-CNN-Combined.

Methods	Acc (%)
AFC [1]	72.8
LR-GCC [2]	75.7
OCB [20]	91.3
ICAC [14]	76.4
BR [56]	86.8
$S^{3}$R [8]	85.3
ResNet-50 [73]	92.4
ResNet-101 [74]	92.3
FGMI-LF-AFC	77.5
FGMI-LF-LR-GCC	78.3
FGMI-LF-OCB	93.6
FGMI-LF-ICAC	79.2
FGMI-LF-BR	89.4
FGMI-LF-$S^{3}$R	88.7
FGMI-CNN-ResNet-50	94.8
FGMI-CNN-ResNet-101	94.2
FGMI-LF-Combined	95.4
FGMI-CNN-Combined	95.9
FGMI-LF-CNN-Combined	**96.5**

**Table 3 sensors-21-04176-t003:** Fine-grained classification performances of FGMI-CNN and other baseline methods on the CUB-200-2011 dataset. FGMI-CNN-AlexNet, FGMI-CNN-VGG, FGMI-CNN-GoogleNet, and FGMI-CNN-BCNN represent the proposed method combined with the prelearned AlexNet, VGG, GoogleNet, and Bilinear CNN models, respectively. FGMI-CNN-Combined represents the proposed method combined with AlexNet, VGG, GoogleNet, and Bilinear CNN jointly. EA—extra annotation.

Methods	EA	Acc (%)	Network
FC-VGG [11]	no	70.4	VGG
bilinear CNN [3]	no	84.1	VGG
LRBP [59]	no	84.2	VGG
WSDL [60]	no	85.7	VGG
PR-CNN [18]	yes	73.5	AlexNet
WS [61]	yes	78.6	AlexNet
PS-CNN [62]	yes	76.2	AlexNet
PN-CNN [63]	yes	75.7	AlexNet
Triplet-A 19	yes	80.7	GoogleNet
STN [64]	no	84.1	GoogleNet
BoostCNN [65]	no	86.2	VGG
HSnet [71]	yes	87.5	GoogLeNet
CVL [72]	yes	85.6	VGG + GoogLeNet
MA-CNN [74]	no	86.5	VGG-19
DFL-CNN [75]	no	87.4	ResNet-50
DCL-VGG-16 [76]	no	86.9	VGG-16
NTS-Net [77]	no	87.5	ResNet-50
DFB-CNN [75]	no	87.4	VGG-16
Cross-X (ResNet) [78]	no	87.7	ResNet
FGMI-CNN-AlexNet	no	73.4	AlexNet
FGMI-CNN-VGG	no	75.8	VGG
FGMI-CNN-GoogleNet	no	83.1	GoogleNet
FGMI-CNN-BCNN	no	86.7	VGG
FGMI-CNN-Combined	no	**88.2**	All

**Table 4 sensors-21-04176-t004:** Fine-grained classification performances of FGMI-CNN and other baseline methods on the Cars-196 dataset.

Methods	EA	Acc (%)	Network
bilinear CNN [3]	no	91.3	VGG
BoostCNN [65]	no	92.1	VGG
BoT [66]	yes	92.5	VGG
FC-VGG [11]	no	76.8	VGG
WSDL [60]	no	92.3	VGG
RCNN [66]	no	57.4	AlexNet
FT-HAR-CNN [67]	no	86.3	AlexNet
MA-CNN [74]	no	92.8	VGG-19
DFL-CNN [75]	no	93.1	ResNet-50
DCL-VGG-16 [76]	no	94.1	VGG-16
FGMI-CNN-AlexNet	no	63.4	AlexNet
FGMI-CNN-VGG	no	84.7	VGG
FGMI-CNN-GoogleNet	no	87.3	GoogleNet
FGMI-CNN-BCNN	no	93.1	VGG
FGMI-CNN-Combined	no	**95.7**	All

**Table 5 sensors-21-04176-t005:** Influences of misclassification information and new image representations on the Flower-102 dataset, CUB-200-2011 dataset, and Cars-196 dataset. no MI—without misclassification information (a simple combination of prelearned models with averaged predictions); no NIR—without new image representation (using the predicted classes over all the selected subsets corresponding to the pretrained models for voting).

Dataset	No MI	No NIR	Proposed Method
Flower-102	93.2	91.6	96.5
CUB-200-2011	85.7	84.5	88.2
Cars-196	93.2	92.4	95.7

## Data Availability

To evaluate the proposed method’s effectiveness (FGMI), we conduct fine-grained image classification experiments on three publicly available datasets: the Flower-102 dataset, the CUB-200-2011 dataset, and the Cars-196 dataset.

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
