# Peer review of "Exploring Misclassification Information for Fine-Grained Image Classification"

_sensors, 2021, doi:10.3390/s21124176_

Round 1

Reviewer 1 Report

This paper proposed a novel fine-grained image classification method by exploring the 5 misclassification information (FGMI) of pre-learned models to make use of misclassification information. But please describe more why this is necessary for the Introduction section.

Please also consider modifying the contents of this paper with the following suggestions.
1. Figure 1 is not clear for the reader to understand the flowchart of the proposed fine-grained image classification method.
2. Algorithm 1 can be easier to understand by giving a real example.
3. Please give more discussions about the results in conclusions, especially, why the proposed fine-grained image classification method is efficient.
4. Please give more descriptions about the meaning of operations and the meaning of mathematics in “FGMI” in section 4.5.

Author Response

Thanks for the valuable reviews. We have revised the paper accordingly.
1. Inspired the review, we have revised Figure 1 to clearly demonstrate the core idea of our proposed model. That is, only the main procedures of exploring the misclassification information with one type of pre-learned model are shown in the figure. The combination procedure that results of several/all types of pre-learned models are integrated to improve the performance is not shown in the figure to avoid confusion. In the beginning of Section 3, we have revised as follows:
We first use the misclassification information from pre-learned models for misclassified image selection. The selected images are then used to train a number of classifiers. We concatenate the outputs of learned classifiers for new image representations which are then used for fine-grained image classifications, as shown in Figure 1. Moreover, to further improve the performance, we design several pre-learned models with different backbones for image representation and classification, and combine these pre-learned models for leveraging the advantages of these models.

2. In Algorithm 1, the procedures mainly involve many classes with a mount of samples, hence we illustrate the Algorithm 1 with the related equations, such as Equation 1-2, 4-5, etc., to show the main procedures, while it is not easy to illustrate the procedures with real examples. However, to make the algorithm clear, we provided more details of the algorithm for demonstration in Section 3.2 as follows:
Algorithm 1 gives the procedures of the proposed fine-grained image classification with misclassification information method. First, the pre-learned classifiers are trained using the training data, and then they are used to predict the images of the training data using Eq. 1. Based on the prediction results, the misclassification information and class distribution can be calculated using Eq. 2. For each class, utilizing the selected $K$ classes that are most likely to be misclassified or confused with the class, the classifier is trained again. The new image representation is obtained by concatenating the result of the $C$ newly trained classifiers, using Eq. 4 and Eq. 5. With the new image representation, the final classifiers are trained by Eq. 6 and Eq. 8. It should be noticed that, using Eq. 6, the classification results can be obtained using one type of pre-learned models, while using Eq. 8, results of several/all types of pre-learned models are combined to obtain the final results, which is expected to improve the performance.

3.In the conclusions, we present more discussions in the revised manuscript, as follows:
In this paper, we proposed an efficient fine-grained image classification method by making use of the misclassification information of pre-learned models, which has been generally ignored by previous methods. We used the learned classifiers to select misclassified images for each class. The selected images were then used to train misclassification classifiers. The selection and training process were conducted for each class. We combined the outputs of these learned classifiers for new image representations and trained classifiers for final predictions. Since the misclassification information contains discriminative features that are important for classification of similar classes with similar semantic and visual appearances. Specifically for the fine-grained classification task, training the classifiers with the misclassification information can better extract confused features that is useful to discriminate similar classes. To evaluate the proposed method's effectiveness, we conducted fine-grained classification experiments on three fine-grained image datasets. Experiential results and analysis proved the effectiveness and usefulness of the proposed method.

4. In Section 4, we have presented the meanings of the evaluated methods, such as FGMI-LF, FGMI-CNN, etc,. We also check carefully about the meanings of the parameters $K$ and \alpha in Section 4.5.There are not many mathematics in this section, but we check carefully about the equations in Section 3. 

Reviewer 2 Report

the paper presents a multi-stage classification process, where a first stage is used to define the objects which are difficult to be classified, and then the misclassified objects are subdivided according to the wrong class attributed to them in the first stage and they are further classified separately. The procedure has been tested both with shallow networks, namely where the features are selected a priori, and with a convolution neural network that instead includes the feature extraction as a part of the training process. As the deep neural networks typically outperform the shallow networks, the obtained improvement is lower in the former case, but it is anyway significant.

The proposed method has been compared with widely referred techniques, showing that  it is always able to achieve better performance.

Remark

The method foresees that only two stage of training are performed, but theoretically nothing prevents to make the procedure more deep, until 100% of the performance is obtained. Is there any theoretical or practical difficulties that make this not convenient?

Minor points:

line 168: "th" instead of "the"

line 189+2: "as£º"

caption of Fig. 4: label of the abscissa has the variable λ not used before

Author Response

Thanks for the valuable comments that mention the possibility of making the procedures deeper. This is a quite good suggestion that deserves further investigation and may need much time and many experiments. It is indeed difficult to determine whether there are theoretical or practical difficulties in doing so. Inspired by the comments, we will further conduct much more experiments and study this interesting research, as our future work.

The minor points have been corrected in the revised version. The variable \lamda should be \alpha. We have revised the figure accordingly.

Reviewer 3 Report

In this study, the authors proposed an efficient fine-grained image classification method by making use of the misclassification information of pre-learned models.  The authors used the learned classifiers to select misclassified images for each class, and the selected images were used to train misclassification classifiers.  The authors combined the outputs of these learned classifiers for new image representations and trained classifiers for final predictions.   The study results look fine.

On Page 4,  the authors defined that " M " is the number of the pre-learned models.  In Tables 2, 3, and 4, what is the numerical value of " M " used for testing the Fine-grained classification performances of the proposed design ?

Author Response

Thanks for the valuable review. In Tables 2, 3, and 4, we list the classification results of each pre-learned model, and then show the combination of them. For example, in Table 2, FGMI-LF-AFC, FGMI-LF-LR-GCC, FGMI-LF-OCB, FGMI-LF-ICAC, FGMI-LF-BR, FGMI-LF-S^3R, FGMI-CNN-ResNet-50 and FGMI-CNN-ResNet-101 represent the proposed method combined with different pre-learned models. FGMI-LF-Combined, FGMI-CNN-Combined, and FGMI-LF-CNN-Combined represent the proposed method combined with a part of the pre-learned models or all of the pre-learned models. For this example, the number M is 8. For Table 3 and Table 4, the number M is 4. In the caption of each table, we have addressed the meanings of these models.